# Uncovering the Effectiveness of Calibration on Open Intent Classification

## Abstract

Open intent classification aims to simultaneously identify known and unknown intents, and it is one of the challenging tasks in modern dialogue systems. While prior approaches are based on known intent classifiers trained under the cross-entropy loss, we presume this loss function yields a representation overly biased to the known intents; thus, it negatively impacts identifying unknown intents. In this study, we propose a novel open intent classification approach that utilizes model calibration into the previously-proposed state-of-the-art. We empirically examine that simply changing a learning objective in a more calibrated manner outperforms the past state-of-the-art. We further excavate that the underlying reason behind calibrated classifier's supremacy derives from the high-level layers of the deep neural networks. We also discover that our approach is robust to harsh settings where few training samples per class exist. Consequentially, we expect our findings and takeaways to exhibit practical guidelines of open intent classification, thus helping to inform future model design choices.

## 1 Introduction

**Background and Motivation** Beyond the success of intent classification under the supervised regime, one of the next challenges in the modern dialogue system is open intent classification (Scheirer et al., 2013). While the number of intents in the training and test sets is the same under the supervised setting (known as a closed-set setting), an intent classifier in the real world is required to recognize unknown intents as well as known intents (Zhang et al., 2021). For example, supposing the training set includes $N$ intents, the open intent classification solves $N + 1$ classification where the $N + 1$ *th* intent is a set of unknown ones (Shu et al., 2017; Lin & Xu, 2019; Zhang et al., 2021). This open intent classification task is also related to open world recognition (Bendale & Boult, 2016; Vaze et al., 2021) or out-of-distribution detection studies (Hendrycks & Gimpel, 2016; Liang et al., 2017) which are actively dealt with image domains, but it is specifically denoted as open intent classification in a natural language processing domain.

Upon previously-proposed open intent classification methods, we figure out that most of these works conventionally trained the closed-set classifier with a cross-entropy loss (Bendale & Boult, 2016; Hendrycks & Gimpel, 2016; Prakhya et al., 2017; Shu et al., 2017; Lin & Xu, 2019; Zhang et al., 2021). *However, we doubt whether this use of cross-entropy loss is the utmost learning objective for identifying open intents.* Previous open intent classification study highlighted that adequate strength of decision boundaries among known intents is important for detecting unknown intents (Zhang et al., 2021). To interpret, an inductive bias established with known intents should be neither overly biased nor too loosely optimized. Not only in open intent classification but recently-proposed state-of-the-art open world classification study in the computer vision domain also supports this proposition: acquiring adequate representation power correlates to effective open world classification performance (Vaze et al., 2021). But, as several works once pointed out, the cross-entropy loss is known to convey an inductive bias that is excessively biased to the given labels because it enforces the model to select one single label among the given label space (Recht et al., 2019; Zhang et al., 2016). To this end, we assume the use of cross-entropy loss has room for improvement and aims to provide an outperforming open intent classifier.

**Main Idea and Its Novelty** Our work's key proposition is utilizing model calibration during the model training on known intents. Model calibration is a method that adjusts a model's predicted

probabilities of outcomes to reflect the true probabilities of those outcomes (Nixon et al., 2019). Referring to the calibration studies, the calibrated deep neural networks accomplished robustness against various noises and perturbations (Müller et al., 2019; Pereyra et al., 2017). Inspired by this finding, we presume that applying calibration to the cross-entropy loss will improve the inductive bias's quality and escalate the open intent classification performance. Accordingly, we select state-of-the-art open-world classification methods in the text and image domains and simply apply calibration to their training procedure. Throughout our work, we firstly showed whether the calibration improves inductive bias compared to the cross-entropy loss. Then, we further examine whether our simple idea can outperform previous open intent classifiers in various problem settings and how calibration changes the representation landscape in the trained model. Although our idea seems to be simple, we highlight that the proposed open intent classifiers are novel because, to the best of our knowledge, our approach is the first attempt to utilize calibration to improve open intent classification performance in the text domain.

**Key Contributions**

- As a preliminary analysis, we show that model calibration reduces the bias of the conventional known intent classifier, as well as escalates the distribution discrepancy between known and unknown intents. We analyze that this large discrepancy would contribute to better open intent classification performances

- We propose two novel methods in open intent classification, C-LC and C-ADB, by applying model calibration to the previously proposed state-of-the-art in image and text domains, respectively. Under the particular settings, we discover the proposed methods become a new state-of-the-art.

- We further scrutinize that the supremacy of C-LC and C-ADB derives from the representations at higher layers of the deep neural networks. We interpret that the proposed methods acquire better contextual understandings than the previously-proposed methods.

- Lastly, we examine our approaches' stability in extreme settings of the training set. We discover that C-ADB is less stable than C-LC given few training samples per known intent; thus, there should be careful consideration on using C-ADB.

## 2 RELATED WORKS

**Open Intent Classification** At first, Scheirer et al. (2013) defined the task of open-set recognition in the computer vision domain and inspired substudies. Fei & Liu (2016) applied SVM with center-based similarity to solve open set classification. Bendale & Boult (2016) preposed OpenMax model, which fits Weibull distribution in the penultimate layer of the network. Prakhya et al. (2017) and Shu et al. (2017) adopt the OpenMax model in open intent classification and show that convolutional neural networks are good feature extractors in NLP domain. Hendrycks & Gimpel (2016) suggest that predicting out-of-distribution example can be distinguished based on softmax probability. Subsequently, post-processing-based methods were proposed. Lin & Xu (2019) apply Large Margin Cosine Loss to the post-processing-based method; the model learns that it maximizes inter-class variance and minimizes intra-class variance. Zhang et al. (2021) introduce learning adaptive decision boundary(ADB) and centroid for open intent classification, which is a post-processing-based method. Shu et al. (2021) use several data augmentation strategies to expand distribution shift examples on ADB. Throughout the prior works, we analyze that a key takeaway of the precise open intent classification method is establishing adequate decision boundaries among the known intent samples; it is also usually denoted as appropriate tightness of decision boundaries (Zhang et al., 2021). As the aforementioned methods commonly employed cross-entropy loss to train the known intent classifier, we hypothesize it is not advantageous in establishing good decision boundaries. Under this motivation, we aim to scrutinize optimal decision boundaries' tightness through applying model calibration to the know intent classifier.

**Model Calibration** Calibration reflects ground truth correctness likelihood in a predicted class label. Good calibrated confidence provides suitable information on why the neural network prediction is made. Guo et al. (2017) proposed temperature scaling to calibrate modern neural networks with over-confident problems. Lee et al. (2017) suggests two additional terms on the original objective function for detecting out-of-distribution. Research of calibration has been widely examined

for computer vision, but natural language processing has only recently begun. Kumar & Sarawagi (2019) presented that neural machine translation models are miscalibrated. Desai & Durrett (2020) tried temperature scaling on BERT and RoBERTa model, and analyzed their calibration over three tasks. In addition, they show that further increasing empirical uncertainty help out-of-domain classification. Moreover, Müller et al. (2019) shows label smoothing behaviors while training network and investigates the effect of label smoothing, which improve model calibration. Among various methods in model calibration, we employed label smoothing in the proposed methods for the following reasons: 1) it does not require any validation set, 2) it directly influences the representation power of deep neural networks while temperature scaling does not.

## 3 OUR APPROACH

This section illustrates detailed descriptions of the proposed open intent classifiers. First, among numerous studies on open-world classification, we selected two state-of-the-art methods for image and text domains. We selected the method proposed in Vaze et al. (2021) from the image domain denoted as Logit-based Classifier (**LC**), whereas Adaptive Decision Boundary (**ADB**) (Zhang et al., 2021) method is employed from the text domain. Note that both approaches utilize cross-entropy loss during the model training. To calibrate these models, we apply label smoothing (Szegedy et al., 2016) with the calibration strength of $\alpha$ and denote them as Calibrated Logit-based Classifier (**C-LC**) and Calibrated ADB (**C-ADB**), respectively. We hereby highlight that temperature scaling approaches (Guo et al., 2017), which are other promising calibration methods, are not considered in our work because they only change absolute logit values without any change in inductive bias. We formalize the cross-entropy loss with label smoothing in equation 1. Note that $K$ implies the number of known intents, $\alpha$ stands for the calibration strength, $p_k$ and $y_k^{LS}$ means logit vector and label-smoothed ground truth, respectively.

$$L_{LS}(p_k, y) = \sum_{k=1}^{K} -y_k^{LS} log(p_k) \; where \; p_k = \frac{e^{x^T w_k}}{\sum_{i=1}^{K} e^{x^T w_i}}, \; y_k^{LS} = y_k(1-\alpha) + \alpha/K \quad (1)$$

**Calibrated Logit-based Classifier (C-LC)** C-LC is an open intent classifier that shares the same motivation with the LC, a recently-proposed open-world classification method (Vaze et al., 2021). Its novelty exists in the calculation of confidence, because it illustrates that simply using the logit vector before the softmax layer can surprisingly increase the open intent classification performance, while most prior works used logit vectors after the softmax layer. During the training procedure, it trains closed-set classifier with cross-entropy loss. Given a test sample, it LC recognizes unknown intents if a given sample's confidence yielded by the trained model is smaller than a preset threshold. Note that this confidence is measured as the maximum value at the logit vector extracted from the layer right before the softmax activation. Lastly, it selects the threshold as a mean confidence at the training samples. On the aforementioend procedures of LC, we establish C-LC by changing the learning objective from simple cross-entropy loss into the calibrated one, shown in Equation 1.

$$L_{C-ADB} = L_{LS} + L_{Boundary} \quad (2)$$

**Calibrated ADB (C-ADB)** C-ADB is another open-intent classifier proposed in our study. The original ADB identifies unknown intents if a given sample locates far from the known clusters' centroids in the latent space. To empower the model to find adequate decision boundaries' tightness, ADB trains the model with cross-entropy loss and boundary loss, where the boundary loss aims to predict the radius of each known intent. We denote this training procedure as post-processing. Given the trained model $\phi(x; \theta)$, each known intent's centroid ($C_1 \cdots C_K$) and radius ($R_1 \cdots R_K$) in the latent space, ADB measures the distance ($d_k$) between given sample's representation and $Kth$ cluster's centroid ($C_k$), and it is denoted as $||\phi(x_{test}; \theta) - C_k||$. The ADB identifies given test sample as known intent if $d_k$ goes smaller than $R_k$. If every $d_k$ goes larger than corresponding $C_k$, it regards it as an unknown intent. We say this open intent classification procedure as post-processing. While the LC simply predicts a given sample as an open intent when its confidence is less than a preset thresdhold, we hereby highlight that ADB utilizes post-proceesor which utilizes estimated distances in the latent space. On the aforementioend establishment procedures of ADB, we calbrate the model with applying label smoothing toward the cross-entropy loss as shown in

Equation 2. We presume calibrated pre-processing procedure and post-processing would make a synergy on identifying unknwn intents.

# 4 EXPERIMENTS

## 4.1 EXPERIMENTAL SETUP

**Dataset and Problem Setting** We utilized three public datasets (STACKOVERFLOW (Xu et al., 2015), BANKING (Casanueva et al., 2020), OOS (Larson et al., 2019)), which are widely utilized in past open intent classification study (Lin & Xu, 2019; Zhang et al., 2021). We show a brief summary of these datasets in Table 1. Upon these datasets, we postulate three problem settings with different Known Label Ratios (KLR). The KLR implies the ratio of known labels to the total number of labels. Supposing the scenario under the KLR of 25%, we use 25% of total intents as known ones while the other 75% intents are set as unknown ones. Thus, the open intent classifier can only learn 25% of the total intents during the training stage and identify both known and unknown intents (75% of total intents) during the test stage. Our study utilized three KLRs of 25%, 50%, and 75% in the experiments. We utilized accuracy and F1-score on the test set as evaluation metrics. Given $N$ intents for the known label, the open intent classifier solves $N + 1$ classification where the added one label implies unknown intents. We report the average performance over five runs of experiments for each known class ratio.

**Implementation and Training Details** Both C-LC and C-ADB utilizes pre-trained BERT provided in Huggingface as a backbone feature extractor as Desai & Durrett (2020) once urged that transformer-based language models are more advantageous in open intent classification. Given a pre-trained BERT, we implement both C-LC and C-ADB by applying calibration to each one's learning objective. For training details, we train these models with batch size of

Table 1: Descriptions of the utilized datasets. Note that the OOS dataset has 150 intents in the training and validation set, but one additional intent exists in the test set.

|  | STACKOVERFLOW | BANKING | OOS |
|---|---|---|---|
| **Number of Intents** | 20 | 77 | 150(+1) |
| **Training Set** | 9003 | 12000 | 15000 |
| **Validation Set** | 1000 | 2000 | 3000 |
| **Test Set** | 3080 | 6000 | 5700 |

128, optimized paramters with AdamW optimizer (Loshchilov & Hutter, 2017). We set the learning rate as 2e-5 and schedule it with cosine scheduiling. We Please refer to the supplementary materials for codes and more details.

**Baselines** We employed five baseline open intent classification methods to examine our approaches' effectiveness. Brief elaborations on the baseline approaches are described as follows. **MSP** Hendrycks & Gimpel (2016) proposed that the out-of-distribution example can be diverged based on the maximum softmax probability when predicting the sample class. **DOC** Shu et al. (2017) applied the CNN layer to open intent classification and $m$ 1-vs-rest sigmoid classifier for $m$ known classes instead of the softmax operation as the final layer. **OpenMax** Bendale & Boult (2016) is a computer vision open-set detection method of distance-based. It applied extreme distribution of distances from the mean in each class to distinguish open set examples. **DeepUnk** Lin & Xu (2019) replaces softmax loss with Large Margin Cosine Loss(LMCL) in a feature extractor model(Bi-LSTM) and feeds these features to the Local Outlier Factor(LOF) for discovering open intent examples. **LC** Vaze et al. (2021) utilizes closed-set classifier's logit values for open intent classification. It claims that a well-trained closed-set classifier's inductive bias is sufficient to identify open-world samples. **ADB** Zhang et al. (2021) proposed the adaptive decision boundary, a post-processing method, to create quality feature representation. Decision boundaries and centroid of each pre-learned class are required for open classification.

## 4.2 DO CALIBRATION REDUCE THE BIAS OF CROSS-ENTROPY LOSS?

**Setup** We first and foremost scrutinize whether the calibration contributes to discriminating unknown intents from the known labels. As we presume that the use of cross-entropy loss yields a large bias to the known intents, we analyze whether the cross-entropy loss indeed exhibits insufficient discriminative decision boundary between known and unknown intents. Moreover, we also aim to validate whether the calibration establishes a more distinct representation between known and unknown intents. To excavate answers to these questions, we trained closed-set intent classifiers

with various calibration options: No calibration (label smoothing with strength $\alpha$ of 0) and label smoothing with the strength of 0.2, 0.5, 0.8, and 0.9. Note that the larger strength implies stronger calibration on the model. Given the trained classifiers, we extract the confidence scores (extracted at the layer before the softmax activation) from known and unknown test samples and analyze their distributions. We regard the more distinct distribution between known and unknown intents casts better representation quality, because it implies the model posits more appropriate discriminative decision boundaries We perform the analyses on three datasets under the 75% KLR, and visualize the confidence distributions in Figure 1. Please refer the supplementary materials for experiment results under the other KLR options.

**Result** Following the results shown in Figure 1, we observe that cross-entropy loss exhibits a particular amount of adjoined area between known and unknown intents. Regardless of dataset types, closed-set classifier poses particular amount of adjoined confidence distributions between known and unknown intents; thus, we conclude that cross-entropy loss bears improvement avenue for more effective open intent classification. We further scrutinize that calibration separates these two distributions by establishing a representation less biased to the known intents. Interestingly, the more strong calibration exhibits a larger discrepancy between known and unknown labels in every KLR. While we could not confidently say the stronger label smoothing creates a better representation for discriminating unknown intents, we resulted that calibration is beneficial to establishing discrepancy between known and unknown intents regardless of KLR levels.

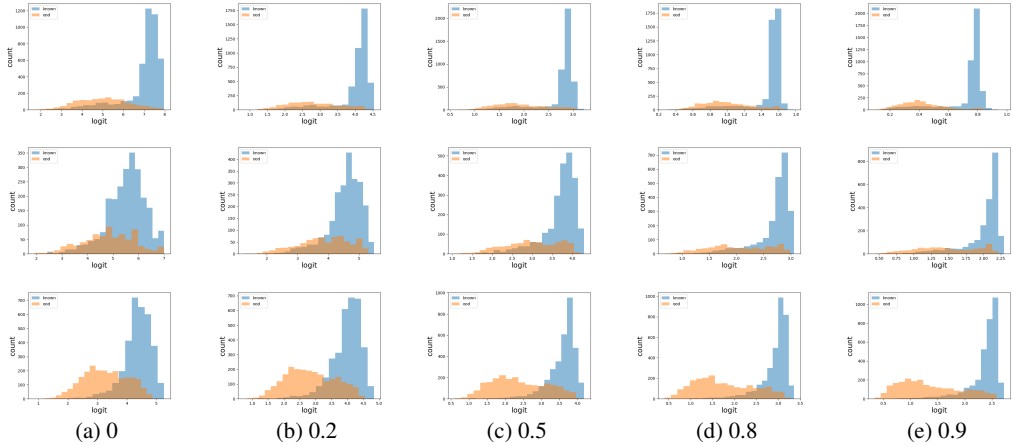

Figure 1: Distribution of maximum value of logit between known intent (blue) and unknown (orange) intent samples under KLR of 75%. From top to bottom, each row indicates the dataset type of STACKOVERFLOW, BANKING, and OOS. From left to right, each column indicates calibration strengths of 0, 0.2, 0.5, 0.8, and 0.9. Note that blue and orange distribution implies known and unknown intents, respectively.

### 4.3 WHAT IS OPTIMAL CALIBRATION STRENGTH?

**Setup** After we confirm the calibration as a presumable solution for the cross-entropy loss's bias, we then excavate what is an optimal calibration strength on C-LC and C-ADB. As the C-ADB has additional post-processing procedure, we assume the adequate calibration strength will be differ from the C-LC's optimal one. To examine this question, we train C-LC and C-ADB along with various calibration strengths, and check the open intent classification performances. We regard an optimal calibration sterngth as the one acheive the best performance. The experiment results are shown in Table 2.

**Result** From Table 2, it shows that each C-LC and C-ADB has different adequate calibration strengths. We interpret this phenomenon highlights the importance of adequate decision boundaries' tightness, that they should not be overly tightened for good open intent classification performance. First, C-LC's performance is maximized at the strongest calibration strength because a closed-set intent classifier without any calibration does not tighten decision boundaries. As conventional cross-entropy loss is insufficient in creating tight decision boundaries, it requires a substantial

impact on distinguishing known intent clusters for good performance; thus, applying the strongest calibration was beneficial in elevating open intent classification performance. On the other hand, C-ADB's performance is maximized at moderate calibration strength. We analyze this phenomenon occurs because the original ADB already bears tightened decision boundaries compared to the LC as the ADB tightens the decision boundaries with post-processing procedure. Suppose we apply the strongest calibration for C-ADB. In this case, we analyze the decision boundaries would become overly tightened as both post-processing procedure and calibration simultaneously tighten the decision boundaries. As the prior study (Zhang et al., 2021) once urged, overly tightened decision boundary degrades the open intent classification performance. We presume a combined influence from post-processing procedure and strong calibration creates extreme tightness of decision boundaries, which leads to inferior performance. Conversely, we presume an adequate tightness of decision boundaries is accomplished by combining post-processor and moderate calibration; therefore, C-ADB with moderate calibration strength achieves the best performance. In a nutshell, we discover that C-LC and C-ADB requires different calibration strenghs for effective open intent classifcation. Furthermore, we confirm that open intent classifier indeed requires an adequate decision boundaries' tightness level and re-assured that overly-tightened decision boundary degrades the performance.

Table 2: Our C-LC and C-ADB's open intent classification performances under various calibration strengths. Each method acquires different calibration strengths for the best performance.

| Method | | C-LC | | | | | | C-ADB | | | | | |
|---|---|---|---|---|---|---|---|---|---|---|---|---|---|
| Dataset | | STACKOVERFLOW | | BANKING | | OOS | | STACKOVERFLOW | | BANKING | | OOS | |
| KLR | Smoothing | Accuracy | F1-score | Accuracy | F1-score | Accuracy | F1-score | Accuracy | F1-score | Accuracy | F1-score | Accuracy | F1-score |
| 25% | 0.2 | 69.29 | 49.20 | 76.93 | 56.68 | 87.19 | 62.49 | 89.99 | 83.33 | 82.54 | 74.48 | 91.11 | 81.04 |
| | 0.5 | 72.41 | 54.05 | 78.89 | 57.78 | 88.18 | 65.11 | **90.07** | **83.85** | **83.58** | **75.46** | **91.65** | **81.45** |
| | 0.8 | 80.80 | 65.65 | 81.15 | 63.18 | 88.82 | 66.93 | 15.45 | 19.77 | 82.02 | 73.46 | 91.52 | 80.82 |
| | 0.9 | **84.55** | **71.12** | **83.69** | **68.59** | **90.12** | **71.81** | 13.89 | 18.61 | 75.71 | 67.63 | 90.98 | 79.52 |
| 50% | 0.2 | 76.12 | 68.04 | 73.05 | 62.76 | 79.42 | 59.72 | 87.09 | **85.51** | 81.52 | **81.43** | **88.23** | **84.61** |
| | 0.5 | 76.61 | 69.01 | 75.79 | 68.61 | 81.45 | 64.98 | **87.10** | 85.47 | **81.66** | 81.14 | **88.23** | 84.15 |
| | 0.8 | 78.37 | 71.33 | 79.08 | 75.31 | 84.19 | 72.70 | 84.76 | 83.04 | 80.88 | 79.94 | 87.97 | 83.54 |
| | 0.9 | **80.50** | **76.18** | **79.98** | **77.34** | **85.52** | **76.55** | 22.35 | 27.60 | 79.79 | 78.38 | 87.84 | 83.33 |
| 75% | 0.2 | 70.40 | 70.34 | 57.56 | 53.49 | 69.68 | 57.01 | 81.90 | 85.14 | **78.82** | **83.66** | **84.60** | **85.18** |
| | 0.5 | 70.46 | 70.17 | 63.47 | 62.59 | 71.95 | 60.54 | **82.24** | **85.41** | 77.66 | 82.64 | 84.06 | 84.40 |
| | 0.8 | 74.06 | 74.80 | 68.72 | 70.56 | 73.92 | 64.64 | 80.70 | 83.99 | 76.36 | 81.40 | 83.70 | 83.85 |
| | 0.9 | **75.25** | **77.90** | **70.45** | **73.27** | **75.81** | **68.75** | 79.20 | 83.04 | 75.13 | 80.29 | 83.90 | 84.06 |

## 4.4 COMPARISON WITH BASELINES

**Setup** After we scrutinize that C-LC and C-ADB requires different calibration strenghs, we then analyze whether the proposed methods outperform previously-proposed approaches. We implement baseline approaches described in section 4.1, and compare the performances in various KLR levels. For the calibration strength of C-LC and C-ADB, we used the ones which accomplish the best accuracy in Table 2 at each KLR under each dataset. The results are shown in Table 3.

**Calibration is advantageous in low KLR** We discover that the proposed methods accomplish promising open intent classification performances compared to the baselines. Especially, the proposed C-ADB outperforms the baselines in KLR of 25% and 50%, while it accomplises competitive performance in 75% KLR. Unfortunately, C-LC achives promising performances but fail to outperform the prior methods. For the C-ADB's competitive performances in 75% KLR, we hypothesize that large KLR setting is more difficult to sensitively discover an optimal calibration strength. The larger number of intents implies larger number of known intent clusters; thus, we expect the model cannot easily establish effective decison boundaries between the known clusters. As Zhang et al. (2021) once urged, especially under the highly-complex decision boundaries, the calibration should have to be carefully applied into the model to maximize the open intent classification. In other words, when many known intents exist, careless calibration has higher risk of demolishing representations on given data. However, we simply applied heuristically-chosen calibration strength (one among 0, 0.2, 0.5, 0.8, 0.9); thus, we expect this careless use of strength is related to the competitive performance of C-ADB.

To experimentally excavate our hypothesis's validity, we visualized the representations yielded from the models at different KLRs. We visualized the representations extracted at the last layer of BERT with t-SNE (Van der Maaten & Hinton, 2008), which is a conventionally utilized method of dimensionality reduction, and the results are shown in Figure 2, 3. These results show that a model under the large KLR exposes less-qualified representation clusters of known intents , which implies that a model's representation is insufficient to discriminate various known intents clearly. Referring to the representation distribution under low KLR, known intents are clearly discriminanted regardless.

Table 3: Comparative study of the proposed methods with baselines

| KLR | Model | STACKOVERFLOW | | BANKING | | OOS | |
|---|---|---|---|---|---|---|---|
| | | Accuracy | F1-score | Accuracy | F1-score | Accuracy | F1-score |
| 25% | MSP | 28.67 | 75.89 | 43.67 | 50.09 | 47.02 | 47.62 |
| | DOC | 42.74 | 76.77 | 56.99 | 58.03 | 74.97 | 66.37 |
| | OpenMax | 40.28 | 77.45 | 49.94 | 54.14 | 68.5 | 61.99 |
| | DeepUnk | 47.84 | 78.52 | 64.21 | 61.36 | 81.43 | 71.16 |
| | LC | 78.09 | 58.61 | 70.86 | 37.18 | 85.00 | 49.25 |
| | ADB | 86.72 | 81.08 | 78.85 | 71.62 | 87.59 | 77.19 |
| | C-LC (OURS) | 84.55 | 71.12 | **83.69** | 68.59 | 90.12 | 71.81 |
| | C-ADB (OURS) | **90.07** | **83.85** | 83.58 | **75.46** | **91.65** | **81.45** |
| 50% | MSP | 52.42 | 63.01 | 59.73 | 71.18 | 62.96 | 70.41 |
| | DOC | 52.53 | 62.84 | 64.81 | 73.12 | 77.16 | 78.26 |
| | OpenMax | 60.35 | 68.18 | 65.31 | 74.24 | 80.11 | 80.56 |
| | DeepUnk | 58.98 | 68.01 | 72.73 | 77.53 | 83.35 | 82.16 |
| | LC | 80.26 | 72.28 | 64.22 | 43.03 | 77.29 | 54.13 |
| | ADB | 86.4 | 85.83 | 78.86 | 80.9 | 86.54 | **85.05** |
| | C-LC (OURS) | 80.50 | 76.18 | 79.98 | 77.34 | 85.52 | 76.55 |
| | C-ADB (OURS) | **87.10** | **85.47** | **81.66** | **81.14** | **88.23** | 84.61 |
| 75% | MSP | 72.17 | 77.95 | 75.89 | 83.6 | 74.07 | 82.38 |
| | DOC | 68.91 | 75.06 | 76.77 | 83.34 | 78.73 | 83.59 |
| | OpenMax | 74.42 | 79.78 | 77.45 | 84.07 | 76.8 | 73.16 |
| | DeepUnk | 72.33 | 78.28 | 78.52 | 84.31 | 83.71 | 86.23 |
| | LC | 71.09 | 70.25 | 52.49 | 46.19 | 66.82 | 52.25 |
| | ADB | **82.78** | **85.99** | 81.08 | **85.96** | 86.32 | 88.53 |
| | C-LC (OURS) | 75.25 | 77.90 | 70.45 | 73.27 | 75.81 | 68.75 |
| | C-ADB (OURS) | 82.24 | 85.41 | 78.82 | 83.66 | 84.60 | 85.18 |

Consequentially, we figure out that calibration is effective in escalating open intent classification performances in low KLRs, but there should be more appropriate manner of selecting the calibration strength. One presumable solution is utilizing a learnable calibration strength for C-LC and C-ADB, but we leave this point as an improvement avenue.

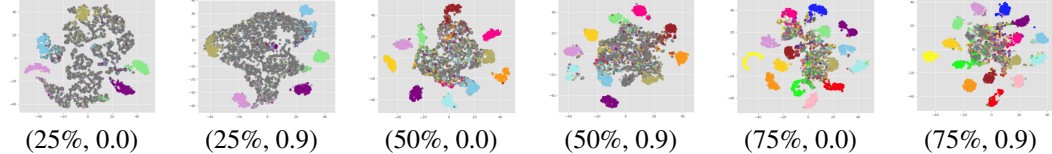

| (25%, 0.0) | (25%, 0.9) | (50%, 0.0) | (50%, 0.9) | (75%, 0.0) | (75%, 0.9) |

Figure 2: The representation distributions between known intent (colored) and unknown intent (gray) samples in STACKOVERFLOW, where the representations are yielded by C-LC. Note that the left element at the bracket implies KLR, and the right one means calibration strength.

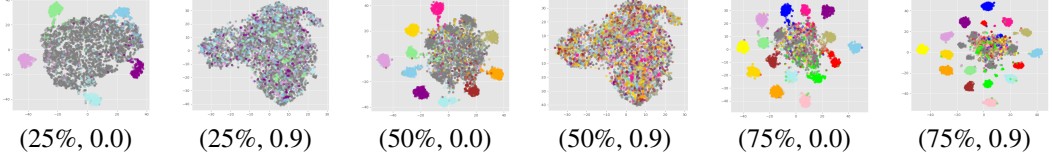

| (25%, 0.0) | (25%, 0.9) | (50%, 0.0) | (50%, 0.9) | (75%, 0.0) | (75%, 0.9) |

Figure 3: The representation distributions between known intent (colored) and unknown intent (gray) samples in STACKOVERFLOW, where the representations are yielded by C-ADB. Note that the left element at the bracket implies KLR, and the right one means calibration strength.

**Validation set can improve C-LC** We figured out that the proposed C-LC could not perform open intent classification better than the original ADB and C-ADB. We analyze ADB-based methods' effectiveness stems from the post-processing procedure as it establishes tighter decision boundaries between known intents. Conversely, in C-LC, the lack of this post-processor might contribute to discover adequate tightness of decision boundaries among known intents; therefore, it could not accomplish comparable performances with ADB-based methods. To improve this limit, we aim to excavate further how we can elevate C-LC's performances. We presume improving a threshold of identifying unknown intents would additionally escalate the performance; thus, we utilized validation sets for choosing this threshold and denoted this method as C-LC-Val. The proposed C-LC-Val uses a validation set consisting of samples from both known and unknown intents. While naive C-LC sets a threshold as

Table 4: Open intent classification performances under the the validation set

| KLR | Model | STACKOVERFLOW | | BANKING | | OOS | |
|---|---|---|---|---|---|---|---|
| | | Accuracy | F1-score | Accuracy | F1-score | Accuracy | F1-score |
| 25% | C-LC | 84.55 | 71.12 | **83.69** | 68.59 | 90.12 | 71.81 |
| | C-ADB | **90.07** | **83.85** | 83.58 | **75.46** | **91.65** | **81.45** |
| | C-LC-Val | 85.56 | 79.06 | 81.38 | 74.04 | 89.48 | 80.78 |
| 50% | C-LC | 80.50 | 76.18 | 79.98 | 77.34 | 85.52 | 76.55 |
| | C-ADB | 87.10 | 85.47 | **81.66** | 81.14 | 88.23 | 84.61 |
| | C-LC-Val | **87.23** | **85.75** | 81.31 | **81.52** | **88.93** | **86.45** |
| 75% | C-LC | 75.25 | 77.90 | 70.45 | 73.27 | 75.81 | 68.75 |
| | C-ADB | 82.24 | 85.41 | **78.82** | **83.66** | 84.60 | 85.18 |
| | C-LC-Val | **85.35** | **87.77** | 78.42 | 81.61 | **85.61** | **85.28** |

mean confidence in the training samples, C-LC-Val chooses a decision threshold set by Youden's J statistics (Youden, 1950) (which is commonly used to set optimal threshold in ROC curve, as well as in prior works (Schisterman et al., 2005; Powers, 2011)). Note that the calibration strength $\alpha$ for C-LC-Val is selected as the one that accomplished the best accuracy in Table 2. We compared the C-LC-Val's performance with other options in Table 4. From Table 4, we scrutinize that C-LC-Val improves naive C-LC's performances in several cases; thus, we conclude that utilizing a validation set can be one presumable method to escalate C-LC's performance. Compared to naive C-LC, We highlight that this assumption of validation set is a limited setting as real-world practitioners cannot always acquire unknown intent samples a priori. But, we aim to show that the practitioners can use C-LC-Val if they can acquire a particular amount of unknown intent samples.

## 4.5 STABILITY UNDER A FEW TRAINING SAMPLES

**Setup** We aim to further excavate whether the proposed methods robustly sustain their performances under the low number of labeled training samples. We define the portion of labeled training samples as Training Data Ratio (TDR), and configure the experiments with TDRs of 0.2, 0.4, 0.6, 0.8, and 1.0. Under the TDR of 0.2, it means there exist only 20% of labeled training samples per each intent. The results are shown in Table 5 and Table 6. Note that $\alpha$ means the calibration strength and a bold number implies the best performance at a given TDR setting.

Table 5: The proposed methods' performances measured in accuracy under various TDR levels.

| KLR | TDR | STACKOVERFLOW C-LC α=0.2 | α=0.5 | α=0.8 | α=0.9 | C-ADB α=0.2 | α=0.5 | α=0.8 | α=0.9 | BANKING C-LC α=0.2 | α=0.5 | α=0.8 | α=0.9 | C-ADB α=0.2 | α=0.5 | α=0.8 | α=0.9 | OOS C-LC α=0.2 | α=0.5 | α=0.8 | α=0.9 | C-ADB α=0.2 | α=0.5 | α=0.8 | α=0.9 |
|---|---|---|---|---|---|---|---|---|---|---|---|---|---|---|---|---|---|---|---|---|---|---|---|---|---|
| 25% | 0.2 | 78.77 | 89.00 | 88.78 | 91.52 | 89.82 | 71.27 | 12.90 | 14.10 | 79.06 | 81.75 | 85.52 | 86.04 | 80.81 | 85.03 | 69.81 | 29.03 | 88.46 | 88.67 | 90.32 | 90.86 | 87.49 | 88.61 | 89.82 | 89.35 |
| | 0.4 | 77.83 | 83.33 | 88.00 | 91.02 | 91.97 | 90.47 | 16.97 | 15.42 | 80.55 | 83.34 | 86.88 | 87.86 | 84.25 | 85.23 | 84.06 | 71.17 | 87.26 | 87.07 | 89.89 | 90.58 | 89.98 | 89.89 | 90.04 | 89.72 |
| | 0.6 | 78.47 | 79.93 | 86.75 | 89.72 | 92.48 | 92.53 | 16.02 | 16.23 | 81.56 | 80.75 | 85.10 | 88.34 | 85.71 | 85.13 | 84.84 | 77.73 | 86.95 | 87.68 | 89.53 | 90.74 | 90.91 | 90.74 | 90.37 | 90.18 |
| | 0.8 | 73.07 | 79.02 | 85.83 | 88.08 | 92.88 | 92.58 | 15.68 | 15.68 | 82.34 | 83.38 | 87.44 | 87.66 | 85.71 | 86.30 | 85.03 | 81.14 | 86.33 | 87.21 | 88.82 | 90.25 | 90.18 | 90.72 | 90.70 | 90.47 |
| | 1 | 72.62 | 73.37 | 81.87 | 86.18 | 92.55 | 93.33 | 16.23 | 16.22 | 80.00 | 82.27 | 85.94 | 87.66 | 86.20 | 85.88 | 86.46 | 79.16 | 85.86 | 87.33 | 88.72 | 90.51 | 90.82 | 90.96 | 90.81 | 90.58 |
| 50% | 0.2 | 85.83 | 79.23 | 80.62 | 83.73 | 88.45 | 88.12 | 85.97 | 19.45 | 68.31 | 74.77 | 76.95 | 75.29 | 74.97 | 76.82 | 76.53 | 74.58 | 79.00 | 80.60 | 82.58 | 83.93 | 85.65 | 85.42 | 84.79 | 83.84 |
| | 0.4 | 84.60 | 82.72 | 82.02 | 84.20 | 88.85 | 88.55 | 87.15 | 19.93 | 72.40 | 76.01 | 78.41 | 78.38 | 79.74 | 79.58 | 80.29 | 78.47 | 79.11 | 80.53 | 83.25 | 84.54 | 86.95 | 87.04 | 86.47 | 86.14 |
| | 0.6 | 82.00 | 84.15 | 83.23 | 83.68 | 88.88 | 88.65 | 87.20 | 23.23 | 69.51 | 71.79 | 76.95 | 78.44 | 81.79 | 81.75 | 81.23 | 79.94 | 78.11 | 80.07 | 83.19 | 83.47 | 87.30 | 87.46 | 87.54 | 87.32 |
| | 0.8 | 82.37 | 80.75 | 79.12 | 83.22 | 88.92 | 88.75 | 87.63 | 23.48 | 70.52 | 73.38 | 78.90 | 79.48 | 81.82 | 82.34 | 81.59 | 79.84 | 78.46 | 80.77 | 83.04 | 83.39 | 87.67 | 87.42 | 87.35 | 87.70 |
| | 1 | 85.35 | 80.58 | 79.77 | 81.43 | 88.87 | 88.85 | 87.23 | 23.07 | 69.58 | 73.12 | 77.27 | 78.51 | 82.95 | 82.79 | 81.85 | 80.55 | 78.89 | 80.49 | 83.12 | 83.75 | 88.00 | 88.07 | 88.19 | 87.54 |
| 75% | 0.2 | 72.62 | 77.95 | 74.43 | 76.88 | 82.47 | 82.35 | 81.73 | 77.63 | 61.98 | 65.45 | 70.26 | 70.00 | 74.71 | 72.53 | 69.68 | 68.41 | 70.77 | 72.95 | 74.30 | 76.95 | 83.12 | 80.86 | 78.84 | 77.54 |
| | 0.4 | 71.77 | 75.32 | 73.08 | 76.58 | 83.02 | 82.98 | 82.38 | 79.67 | 58.96 | 62.05 | 66.30 | 69.58 | 78.25 | 77.21 | 75.62 | 74.45 | 71.26 | 72.96 | 74.11 | 76.81 | 84.21 | 82.58 | 81.39 | 80.74 |
| | 0.6 | 74.98 | 74.48 | 74.03 | 77.83 | 83.27 | 83.27 | 82.32 | 82.72 | 60.52 | 66.56 | 69.74 | 72.31 | 80.19 | 79.06 | 77.08 | 76.07 | 71.14 | 73.26 | 73.05 | 76.33 | 83.93 | 82.98 | 82.65 | 82.23 |
| | 0.8 | 72.95 | 73.17 | 74.83 | 75.25 | 83.65 | 83.62 | 82.57 | 81.50 | 60.03 | 66.23 | 66.85 | 69.45 | 81.66 | 79.84 | 78.51 | 77.27 | 72.04 | 73.16 | 73.56 | 75.84 | 83.93 | 83.33 | 83.30 | 83.19 |
| | 1 | 70.97 | 72.95 | 74.20 | 78.17 | 83.62 | 83.77 | 82.42 | 81.98 | 60.19 | 67.01 | 68.86 | 71.23 | 81.66 | 80.42 | 78.93 | 78.47 | 71.47 | 72.56 | 73.02 | 75.26 | 84.40 | 83.86 | 83.56 | 83.91 |

Table 6: The proposed methods' performances measured in F1 score under various TDR levels.

| KLR | TDR | STACKOVERFLOW C-LC α=0.2 | α=0.5 | α=0.8 | α=0.9 | C-ADB α=0.2 | α=0.5 | α=0.8 | α=0.9 | BANKING C-LC α=0.2 | α=0.5 | α=0.8 | α=0.9 | C-ADB α=0.2 | α=0.5 | α=0.8 | α=0.9 | OOS C-LC α=0.2 | α=0.5 | α=0.8 | α=0.9 | C-ADB α=0.2 | α=0.5 | α=0.8 | α=0.9 |
|---|---|---|---|---|---|---|---|---|---|---|---|---|---|---|---|---|---|---|---|---|---|---|---|---|---|
| 25% | 0.2 | 63.09 | 82.69 | 75.43 | 81.37 | 83.00 | 70.82 | 17.55 | 18.11 | 64.19 | 68.62 | 72.75 | 71.82 | 68.13 | 71.82 | 60.81 | 39.92 | 65.83 | 66.97 | 75.11 | 76.26 | 71.52 | 72.33 | 72.81 | 71.93 |
| | 0.4 | 61.27 | 76.25 | 79.93 | 82.76 | 85.74 | 84.23 | 20.53 | 19.92 | 68.40 | 68.38 | 74.64 | 77.55 | 74.16 | 74.80 | 72.64 | 63.94 | 63.50 | 62.19 | 75.27 | 77.40 | 76.35 | 75.13 | 74.59 | 74.71 |
| | 0.6 | 74.72 | 71.83 | 72.42 | 78.88 | 86.45 | 86.54 | 21.02 | 21.07 | 71.91 | 62.47 | 68.43 | 79.34 | 77.22 | 75.95 | 75.34 | 69.76 | 63.02 | 64.86 | 73.51 | 78.43 | 78.95 | 78.05 | 76.40 | 76.18 |
| | 0.8 | 58.91 | 64.07 | 75.11 | 74.69 | 87.01 | 86.54 | 22.52 | 22.47 | 70.74 | 69.86 | 74.64 | 77.47 | 77.14 | 78.55 | 75.57 | 71.83 | 58.51 | 63.41 | 69.29 | 74.91 | 77.99 | 78.73 | 78.23 | 77.51 |
| | 1 | 52.16 | 48.23 | 65.93 | 71.10 | 86.54 | 87.69 | 21.71 | 21.66 | 67.13 | 63.42 | 70.27 | 76.77 | 78.39 | 78.04 | 78.09 | 70.17 | 56.54 | 62.82 | 69.48 | 75.83 | 79.36 | 79.35 | 78.37 | 77.56 |
| 50% | 0.2 | 82.43 | 69.19 | 72.28 | 78.65 | 87.12 | 86.66 | 84.93 | 24.69 | 56.72 | 68.13 | 71.20 | 69.63 | 72.42 | 73.09 | 71.84 | 70.47 | 59.72 | 65.01 | 70.47 | 74.22 | 79.82 | 78.50 | 77.21 | 75.49 |
| | 0.4 | 80.60 | 76.62 | 74.46 | 80.33 | 87.54 | 87.18 | 85.76 | 26.93 | 60.76 | 67.17 | 72.09 | 72.98 | 78.84 | 77.88 | 77.67 | 75.79 | 59.02 | 63.61 | 71.10 | 74.86 | 82.34 | 81.69 | 80.37 | 79.65 |
| | 0.6 | 76.06 | 78.50 | 76.92 | 80.33 | 87.59 | 87.28 | 85.48 | 29.55 | 55.66 | 62.10 | 70.30 | 73.50 | 80.85 | 80.26 | 79.08 | 77.30 | 56.87 | 61.40 | 70.59 | 71.78 | 82.91 | 82.52 | 82.26 | 81.76 |
| | 0.8 | 77.70 | 74.58 | 70.18 | 78.43 | 87.64 | 87.39 | 86.02 | 29.28 | 56.04 | 61.99 | 72.93 | 74.28 | 81.20 | 81.22 | 79.90 | 77.83 | 57.82 | 63.64 | 70.55 | 71.68 | 83.34 | 82.68 | 82.14 | 82.70 |
| | 1 | 83.25 | 74.35 | 70.77 | 75.50 | 87.63 | 87.54 | 85.66 | 29.51 | 55.15 | 64.79 | 72.75 | 74.00 | 82.52 | 82.06 | 80.54 | 78.69 | 59.34 | 63.57 | 70.82 | 72.50 | 83.77 | 83.60 | 83.59 | 82.78 |
| 75% | 0.2 | 73.70 | 78.89 | 73.48 | 79.83 | 85.77 | 85.64 | 85.09 | 82.83 | 59.54 | 64.29 | 71.65 | 71.45 | 78.82 | 76.73 | 73.61 | 71.69 | 58.05 | 62.59 | 66.06 | 71.82 | 83.24 | 80.30 | 77.70 | 75.98 |
| | 0.4 | 70.46 | 73.54 | 70.89 | 79.05 | 86.24 | 86.19 | 85.61 | 83.95 | 50.96 | 57.17 | 64.94 | 69.91 | 82.30 | 81.29 | 79.57 | 78.38 | 58.37 | 61.98 | 64.46 | 69.94 | 84.57 | 82.29 | 80.64 | 79.85 |
| | 0.6 | 74.09 | 72.45 | 71.73 | 80.81 | 86.46 | 86.44 | 85.58 | 86.08 | 53.52 | 64.17 | 69.83 | 73.06 | 84.47 | 83.32 | 81.29 | 80.09 | 58.13 | 61.69 | 62.08 | 68.74 | 84.16 | 82.74 | 82.24 | 81.79 |
| | 0.8 | 71.58 | 71.03 | 73.55 | 77.20 | 86.78 | 86.73 | 85.76 | 85.22 | 51.94 | 63.41 | 65.09 | 69.04 | 85.62 | 84.02 | 82.63 | 81.39 | 59.94 | 61.54 | 62.73 | 67.40 | 84.10 | 83.18 | 83.03 | 82.86 |
| | 1 | 69.82 | 71.12 | 73.21 | 80.72 | 86.75 | 86.87 | 85.60 | 85.59 | 53.50 | 64.81 | 68.13 | 71.88 | 85.58 | 84.57 | 83.11 | 82.66 | 59.45 | 60.87 | 61.95 | 66.63 | 84.84 | 84.04 | 83.46 | 83.89 |

**Result** We discover that key findings presented in the section 4.4 still exist unless there are fewer training samples per class. Nevertheless, we observe that C-ADB is less stable than C-LC under the various training samples per class; thus, we analyze that there should be a careful configuration of C-ADB, especially under a few training samples. Following the results, C-LC's performances do not significantly change under various TDR levels, while C-ADB's performances bear such high variability. Particularly, C-ADB's performance experiences a harsh drop under low KLR (i.e., 25%) and high calibration strength. We presume the underlying reasons behind these phenomena are also the tightness of decision boundaries. Supposing low TDR, a model would learn insufficient knowledge during the training; thus, we expect it to risk overfitting and disqualified understanding on known intents. Under the overfitted, disqualified representations, we presume applying calibration or post-processors (which escalated open intent classification performance under large training samples) would not contribute to acquiring better decision boundaries. Accordingly, it degrades the open intent classification performance. Therefore, we urge that applying calibrations or post-processor is not a golden key to escalating performance in every circumstance. Still, we also discover that the landscape of performance change along with various KLR and TDR levels depends on the dataset. We yield this analysis as an improvement avenue as it is not a core component of our study. As a naive direction of analysis, we expect there should be a method of quantifying the decision boundaries' tightness for the open intent classification task. In a nutshell, we recommend

that NLP practitioners use the proposed C-ADB carefully. We highlight that C-ADB might yield inconsistent performance if a few training samples exist on known intents.

## 5 WHY DO CALIBRATION CONTRIBUTE TO OPEN INTENT CLASSIFICATION?

**Setup** Lastly, we aim to excavate how the model calibration contributes to the escalation of open intent classification performances. We hypothesize the answer would exist in the learned representations of the model; thus, we measured representation similarities between various models trained under different calibration strengths. We establish five pairs consisting of models trained under different calibration strength as follows: (0 v. 0), (0 v. 0.2), (0 v.0.5), (0 v. 0.8) and (0 v. 0.9). Given these pairs, we quantitatively measure the similarity between two models in a pair by applying Centered Kernel Alignment (CKA) (Kornblith et al., 2019). CKA measures representation similarity between two layers from different models and returns the similarity score between 0 and 1 (where 1 means the highest similarity). Among various methods in quantifying representation similarities, such as CCA or SVCCA, we utilize CKA as it accomplished state-of-the-art performances in their domain's benchmarks (Kornblith et al., 2019; Wu et al., 2020; Sridhar & Sarah, 2020). Following the prior work (Kornblith et al., 2019), we compared representation similarities at every LayerNorm layer between two models in a given pair. Due to the page limits, we show representation similarities among various models trained under STACKOVERFLOW dataset in Figure 4. Please refer to the supplementary materials for representation similarities on the other datasets.

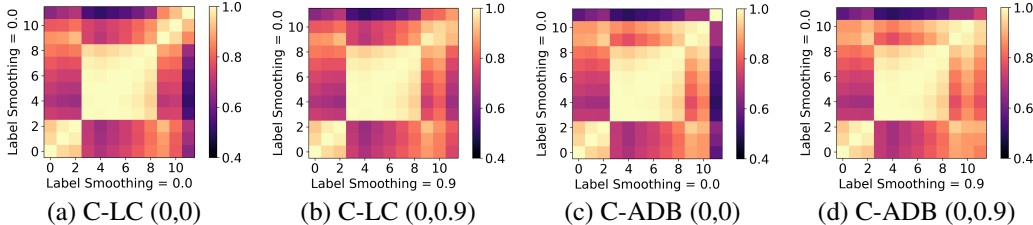

| (a) C-LC (0,0) | (b) C-LC (0,0.9) | (c) C-ADB (0,0) | (d) C-ADB (0,0.9) |

Figure 4: Representation similarities between models with and without calibration

**Result** Upon the Figure 4, for both C-LC and C-ADB, we scrutinize that model calibration yields different high-level representations from the model without any calibration. While overall landscapes of representation similarities look similar regardless of calibration strengths, similarity at the very last layer (11th layer) becomes less similar as long as the calibration strength increases. We interpret this phenomenon implies that calibration makes the model's high-level representation becomes different. Along with prior work's analysis (Hao et al., 2019; Wu et al., 2020), we further hypothesize that calibrated model bears a different contextual understanding of given text input. We analyze calibrated model interprets the text input as less biased to the known intents; thus, it discriminates unknown intents based on this qualified understanding of the text data. Note that this analysis supports that calibrated model acquires 'different' representation from the not-calibrated one, but it does not justify how the calibrated representation yields better open intent classification performance. We let this point as an improvement avenue.

## 6 CONCLUSION

In this study, we propose novel open intent classification methods that utilize label smoothing on prior state-of-the-art methods. We experimentally show this simple idea improves prior approaches' performance in particular settings as calibrated representation makes an adequate tightness on decision boundaries among known intents; thus, the proposed C-ADB becomes a novel state-of-the-art in benchmark settings. Furthermore, we analyze the proposed methods' supremacy derives from high-level representations, which implies model calibration contributes to acquiring a more qualified contextual understanding of text inputs. Last but not least, we also scrutinize that C-ADB is less stable than C-LC under a few training samples; thus, we highly recommend that practitioners carefully utilize it on their own datasets. Nevertheless, several improvement avenues exist as proposed in the prior sections. We expect practitioners in real world can use the proposed methods, especially C-ADB, to establish effective NLP applications in their domains.

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
