# OpenReview forum: "Uncovering the Effectiveness of Calibration on Open Intent Classification"
_ICLR.cc/2023/Conference — Submitted to ICLR 2023_

### Official Review · Reviewer_UJRs · 2022-10-24

**Confidence:** 3
**Correctness:** 3
**Technical Novelty And Significance:** 2
**Empirical Novelty And Significance:** 3
**Recommendation:** 3

**Clarity, Quality, Novelty And Reproducibility:**

The clarity is good. The quality and novelty are not fair enough. The experiments seem reproducible.

**Strength And Weaknesses:**

Strength:

The main idea of the paper is clear. The experimental part is solid with sufficient empirical results and analyses.

Weaknesses:

About the Novelty: the proposed method is more like an empirical trick or a modification to the previous baseline LC and ADB. The motivation and justification to use the label smoothing trick remain unclear.

About the Experiments: The proposed methods did not significantly outperform the previous baseline. On the OOS dataset, the proposed C-ADB even got worse performance than the original ADB, which makes the effectiveness of the proposed method unconvincing.

**Summary Of The Paper:**

The paper proposes two open intent classification methods based on the label-smoothing method, C-LC, and C-ADB. More specifically, the label smoothing method takes a weighted average of the ground-true label and the uniform 1/k vector. The experiments
evaluates the proposed methods on real-world datasets with previous baselines.

**Summary Of The Review:**

The method described in the paper is more like an empirical trick than a well-designed method. The motivation and justification for the method are not clearly stated. The experimental results did not show significant improvement when applying the proposed method.

---

> ### Author Response · Authors · 2022-11-13
> **Response to Reviewer UJRs**
>
> We appreciate your detailed comments and hereby carefully address each of your concerns as follows. We also updated our manuscript according to the suggestions.
>
> **W1. About the Novelty: the proposed method is more like an empirical trick or a modification to the previous baseline LC and ADB. The motivation and justification to use the label smoothing trick remain unclear.**
>
> Thank you for your comment. **We respectfully highlight that we already explained the motivation and justification for using label smoothing for open intent classification in the Introduction section.**
>
> To briefly summarize, we first discover that most of the recent open intent classification tasks conventionally trained closed-set classifiers with cross-entropy loss [1-5]. However, we wondered whether using the cross-entropy loss is the optimal learning objective for open set recognition. Furthermore, a previous study [5] proposed that adequate strength of decision boundaries among known intents is critical for detecting unknown intent examples. Therefore, it is interpreted that the indutive bias established with known intents should be neither overly biased nor too loosely optimized. Based on this motivation, we apply model calibration to the cross-entropy loss during the model training on known intents, and we presume it would improve the model's inductive bias for better open intent classification.
>
> As a justification of our hypothesis, throughout the experiment results shown in our work, we showed our hypothesis was a valid one. We also observe that our approach accomplishes better open intent classification performances, especially under extreme circumstances (where a few samples exist per class).
>
> Consequentially, we respectfully highlight our work's motivation and justification for utilizing calibration onto the open intent classification.
>
> [1] Abhijit Bendale and Terrance E Boult. Towards open set deep networks. In Proceedings of the IEEE conference on computer vision and pattern recognition, pp. 1563–1572, 2016.
>
> [2] Sridhama Prakhya, Vinodini Venkataram, and Jugal Kalita. Open set text classification using CNNs. In Proceedings of the 14th International Conference on Natural Language Processing (ICON2017), pp. 466–475, Kolkata, India, December 2017. NLP Association of India. URL https: //aclanthology.org/W17-7557.
>
> [3] Lei Shu, Hu Xu, and Bing Liu. Doc: Deep open classification of text documents. arXiv preprint arXiv:1709.08716, 2017.
>
> [4] Ting-En Lin and Hua Xu. Deep unknown intent detection with margin loss. arXiv preprint arXiv:1906.00434, 2019.
>
> [5] Hanlei Zhang, Hua Xu, and Ting-En Lin. Deep open intent classification with adaptive decision boundary. In Proceedings of the AAAI Conference on Artificial Intelligence, volume 35, pp. 14374–14382, 2021.
>
> **W2. About the Experiments: The proposed methods did not significantly outperform the previous baseline. On the OOS dataset, the proposed C-ADB even got worse performance than the original ADB, which makes the effectiveness of the proposed method unconvincing.**
>
> Thank you for raising your concern. We fully understand the reviewer might doubt the C-ADB's performance.
>
> **We would like to highlight our method's effectiveness under extreme circumstances where a few training samples exist per class.** While every baseline experiences harsh performance drops under extreme circumstances, our methods consistently achieve precise open intent classification performances. While our method could not outperform the prior baseline in particular cases (where many training samples exist per class), we hope the reviewer focus on our method's effectiveness in extreme settings. We presume the NLP community cannot always guarantee many training samples per class in the real world. We believe our takeaways can be a useful message to the community to solve various open intent classification tasks under extreme settings.
>
> **Moreover, we respectfully highlight the proposed method's performance (especially, C-ADB) achieved promising performances with a small gap from ADB.** Referring to Tabel 3, under KLR of 75%, our C-ADB accomplished a performance distinct from the ADB under 2-3%. We fully acknowledge that our method could not outperform the prior state-of-the-art only in 75% of KLR settings; however, considering the aforementioned takeaway, we presume the effectiveness of calibration on open intent classification is sufficiently supported.
>
> Last but not least, as the reviewer pointed out, we strongly agree that our method can be surely improved to achieve better performance. To mitigate this issue, We believe that the calibration strength can be parameterized as part of the training operation and can serve as an important improvement in future works.
>
> We thank you again for your time and efforts in reviewing our paper and the constructive comments and suggestions.

---

### Official Review · Reviewer_HHjh · 2022-10-24

**Confidence:** 3
**Correctness:** 3
**Technical Novelty And Significance:** 2
**Empirical Novelty And Significance:** 2
**Recommendation:** 5

**Clarity, Quality, Novelty And Reproducibility:**

This paper presents an interesting dimension to investigate the benefits of model calibration in open intent classification problem. For establishing concrete results, at least one other calibration method should be used for drawing the conclusive evidence that indeed model calibration incorporates the inductive bias into open intent classification setup. One more out of domain intent classification method can be used in addition to ADB and LC.

**Strength And Weaknesses:**

Strengths

S1: It is an interesting idea to utilize model calibration towards open intent classification.

S2: The proposed approach has been applied to text and image domain state-of-the-art methods.

S3: The paper presents a detailed set of experiments.

S4: The paper is easy to follow, but writing needs improvement.


Weaknesses

W1: The proposed approach has been empirically verified on two state-of-the-art methods, which are Adaptive Decision Boundary (ADB) and Logit-based classifier (LC). It could have been interesting to see if the approach generalizes to a few others out of domain intent classification methods.
W2: It will be interesting to investigate the behavior of other calibration methods like temperature scaling with the proposed approach. This will add the element of completeness in the research question being investigated.

W3: Logit-based classifier should be one of the baselines.

W4: There are various grammatical mistakes and typos in the paper.  i) For example, in section 4.3, “we assume the adequate calibration strength will be differ from the C-LC’s optimal one.”  ii) Similarly, in section 4.2 in results section, “we resulted that calibration itself is beneficial”. iii) Typo ⇒ “optinmal calibration sterngth”

W5: Figure 1 has enough empty space which can be utilized for increasing the subplot sizes. Especially, legend is difficult to read.


**Summary Of The Paper:**

This paper studies the open intent classification problem. There are standard supervised learning approaches for this task, but the authors hypothesize that cross entropy based objective in the existing open intent classification problem leads to predictions which are overly biased towards known intents, leading to poor performance on open intents. It has been presented with empirical evidence that changing a learning objective in a more calibrated manner can lead to better performance and outperforms the existing state-of-the-art. Authors also investigate the reason for the supremacy of the calibrated model and highlight its connection with higher layers in the model.

**Summary Of The Review:**

Overall, this paper investigates an interesting question. For the thorough investigation and establishing the claims, more experimentation is required.

---

> ### Author Response · Authors · 2022-11-13
> **Respone to Reviewer HHjh**
>
> We appreciate your detailed comments and hereby carefully address each of your concerns as follows. We also updated our manuscript according to the suggestions.
>
> **W1: The proposed approach has been empirically verified on two state-of-the-art methods, which are Adaptive Decision Boundary (ADB) and Logit-based classifier (LC). It could have been interesting to see if the approach generalizes to a few others out of domain intent classification methods.**
>
> Thank you for your suggestion. We would like to highlight that we have already compared our method's performance with various Out-Of-Domain Intent classification methods. Referring to Table 3, we report comparative experiment results with **MSP**, **DOC**, **OpenMax**, which are widely-utilized baselines for both open intent classification and out-of-distribution detection studies. We also highlight that our methods accomplished improved open intent classification performance compared to these previous OOD methods.
>
> **W2: It will be interesting to investigate the behavior of other calibration methods like temperature scaling with the proposed approach. This will add the element of completeness in the research question being investigated.**
>
> Thank you for your valuable comment. We respectfully emphasize that temperature scaling does not fit in the open intent classification task, as it does not change the model's inductive bias. The temperature scaling aims to achieve calibration by rescaling the absolute logit value by adding a single scalar parameter. So, the temperature scaling does not change the model's inductive bias as well as representation quality; instead, it only changes the logit vector's absolute value yielded by the model. When we ideate our work, we also considered using temperature scaling as a model calibration method. But, we dropped it due to the aforementioned reason.
>
> Last but not least, please note that we already highlighted the reason why we did not consider temperature scaling for calibration in Section 3. Please search the sentence starting with "We hereby highlight that temperature scaling approaches ...".
>
> **W3: Logit-based classifier should be one of the baselines.**
>
> We strongly agree with the reviewer's comment, and we performed additional experiments on the public benchmark with a Logit-based classifier (LC). We added additional descriptions on the LC in Section 4.1 (please check the baseline subsection in the revised manuscript), and we report the open intent classification performances in Table 4. We discover that our method accomplishes improved performances compared to the LC; therefore, we hereby highlight that our work's effectiveness is still valid.
>
> Again, we sincerely appreciate the reviewer for pointing out this important issue. With your valuable comment, we expect our work's contribution and key takeaways will be more clearly delivered to the readers.
>
>
> **W4: There are various grammatical mistakes and typos in the paper**
>
> We appreciate the reviewer commenting on grammatical errors and typos. Note that we revised the comments that the reviewer pointed out as follows, and we highlighted them in yellow in the revised manuscript.
> - In section 4.3, we changed "be differ" to "differ".
> - In section 4.2, we changed "we resulted that calibration itself is beneficial" to "we resulted that calibration is beneficial".
> - In section 4.3, we changed "optinmal" to "optimal".
>
> **W5: Figure 1 has enough empty space which can be utilized for increasing the subplot sizes. Especially, legend is difficult to read.**
>
> We appreciate the reviewer's comment on the figure size. We increased the size of the subplot as much as possible and reflected it in the manuscript. Please refer to Figure 1 in the revised manuscript.
>
> We thank you again for your time and efforts in reviewing our paper and the constructive comments and suggestions. We hope you will consider raising your score if you find our response satisfactory.

---

### Official Review · Reviewer_HPLr · 2022-10-27

**Confidence:** 4
**Clarity, Quality, Novelty And Reproducibility:** see above.
**Correctness:** 3
**Technical Novelty And Significance:** 2
**Empirical Novelty And Significance:** 3
**Recommendation:** 3

**Strength And Weaknesses:**

**Strengths**
1. The proposed method is clear and simple;
2. The analysis and experiments are somewhat comprehensive.

**Weaknesses**
1. The authors claim the benefit of calibration by the proposed method, but fail to show an evaluation of the calibration performance in terms of expected calibration error (ECE).

2. The experiment evaluation should be further strengthened. The following related prior works are missing and should be compared.

a. Likelihood ratios and generative classifiers for unsupervised out-of-domain detection in task oriented dialog, AAAI 2020

b. Joint Energy-based Model Training for Better Calibrated Natural Language Understanding Models, EACL 2021

**Summary Of The Paper:**

In general, this paper proposes to use the label smoothing method to calibrate the model in open intent classification problem, based on two previously used method logit classifier (LC) and adaptive decision boundary (ADB). The result in low KLR (Known Label Ratios) shows the superiority of the proposed method over previous methods, while the result in high KLR does not. Extensive experiments are taken to demonstrate that the proposed method can achieve improvements in inductive bias, representation landscape and open intent classifier performance.

**Summary Of The Review:**

see above.

---

> ### Author Response · Authors · 2022-11-13
> **Response to Reviewer HPLr**
>
> We appreciate your detailed comments and hereby carefully address each of your concerns as follows. We hope you consider raising your score if you find our response satisfactory.
>
> **Q1. The authors claim the benefit of calibration by the proposed method, but fail to show an evaluation of the calibration performance in terms of expected calibration error (ECE).**
>
> Thank you for your suggestion. We hereby highlight that calibration performance is out of our work's scope; instead, we focus on open intent classification performance. As we denoted in the introduction, our key motivation is that model calibration would improve the model's inductive bias as well as open intent classification performance. We emphasize that raising the calibration performance (measured in ECE) is not our work's analysis scope; therefore, we omitted ECE in the manuscript.
>
> **Q2. The experiment evaluation should be further strengthened. The following related prior works are missing and should be compared.**
>
> We appreciate the reviewer for pointing out these valuable works.
>
> For paper [1], we agree that a comparative evaluation of our methods with the one proposed in [1] could benefit the readers. However, we respectfully urge that the other baseline (i.e., ADB) is stronger baseline than [1] as it achieved better open intent classification performances in various public benchmarks. Therefore, our comparative experiment results with ADB can alternate the results shown in [1].
>
> For paper [2], while the method proposed in [2] shares similar motivation with our work, we hereby highlight that the target task differs from open intent classification; Thus, we respectfully urge that comparative evaluation does not provide useful takeaways to the readers. The proposed method in [2] aims to escalate the model's inductive bias on general NLU tasks, not the open intent classification task. We urge that an inductive bias designed for general tasks would not fit with open intent classification tasks, as many works have shown it was ineffective in prior studies. Therefore, we suggest that comparative evaluation with [2] does not fit with our work.
>
> [1] Likelihood ratios and generative classifiers for unsupervised out-of-domain detection in task oriented dialog, AAAI 2020
>
> [2] Joint Energy-based Model Training for Better Calibrated Natural Language Understanding Models, EACL 2021
>
> We thank you again for your time and efforts in reviewing our paper and the constructive comments and suggestions. We hope that you will consider raising your score if you find our response satisfactory.

---

> > ### Comment · Reviewer_HPLr · 2022-12-01
> > **After reading the response from the authors**
> >
> > Thanks for your response. Unfortunately, the response does not address my concerns.
> >
> > For Q1: just saying "raising the calibration performance (measured in ECE) is not our work's analysis scope" does not remove the necessity for such study and comparison.
> >
> > For Q2:
> >
> > >"the other baseline (i.e., ADB) is stronger baseline than [1] as it achieved better open intent classification performances in various public benchmarks."
> >
> > Is there any literature showing that ADB outperforms the methods in [1]?
> >
> > >"the target task differs from open intent classification"
> >
> > What you need to compare is different methods, which may be applied in different task in the original papers.

---

### Author Response · Authors · 2022-11-13
**Response to All Reviewers**

We sincerely appreciate every reviewer for valuable comments on our work. We are glad that the reviewers generally find our paper written with meaningful contributions to the research community.

We addressed your comments regarding our work in your review sections. Please have a look, and let us know if there are any points of clarification or additional concerns about our work or responses. We will follow up as soon as possible for your decision. We hope you raise your score if our responses are satisfactory.

We sincerely hope that this work will benefit the communities of machine learning and NLP. We have made some changes to address the comments from the reviewers. We would really appreciate it if the reviewers checked our revised manuscript.

---

### Decision · Program_Chairs · 2023-01-20

**Decision:**

Reject

**Justification For Why Not Higher Score:**

A more complete set of baselines and comparisons (or more compelling arguments of why such comparisons are not provided) would be a necessary step to obtain a higher score.

**Justification For Why Not Lower Score:**

N/A

**Metareview: Summary, Strengths And Weaknesses:**

The paper presents an approach to open intent classification based on calibration measures. There is an agreement that the paper presents the ideas clearly, and that the problem addressed is interesting. However, there is also an agreement that the weaknesses outweight the strengths of the paper.

Thee is a consensus among reviewers that the main weaknesses stem from a limited novelty and missing comparisons. While the rebuttal provided some promising elements of response, the main questions remain (see e.g., Rev HPLr's reply).